# Ultrasensitive Stress Biomarker Detection Using Polypyrrole Nanotube Coupled to a Field-Effect Transistor

**DOI:** 10.3390/mi11040439

**Published:** 2020-04-22

**Authors:** Kyung Ho Kim, Sang Hun Lee, Sung Eun Seo, Joonwon Bae, Seon Joo Park, Oh Seok Kwon

**Affiliations:** 1Infectious disease Research Center, Korea Research Institute of Bioscience and Biotechnology (KRIBB), Daejeon 34141, Korea; doublekh0119@gmail.com (K.H.K.); eun93618@kribb.re.kr (S.E.S.); 2Department of Chemical Engineering and Applied Chemistry, Chungnam National University, Daejeon 305-764, Korea; 3Department of Bioengineering, University of California Berkeley, Berkeley, CA 94720, USA; shlee.ucb@gmail.com; 4Department of Applied Chemistry, Dongduk Women’s University, Seoul 02748, Korea; redsox7@dongduk.ac.kr

**Keywords:** cortisol, stress hormone, conducting polymer, polypyrrole nanotube, field-effect transistor

## Abstract

Stress biomarkers such as hormones and neurotransmitters in bodily fluids can indicate an individual’s physical and mental state, as well as influence their quality of life and health. Thus, sensitive and rapid detection of stress biomarkers (e.g., cortisol) is important for management of various diseases with harmful symptoms, including post-traumatic stress disorder and depression. Here, we describe rapid and sensitive cortisol detection based on a conducting polymer (CP) nanotube (NT) field-effect transistor (FET) platform. The synthesized polypyrrole (PPy) NT was functionalized with the cortisol antibody immunoglobulin G (IgG) for the sensitive and specific detection of cortisol hormone. The anti-cortisol IgG was covalently attached to a basal plane of PPy NT through an amide bond between the carboxyl group of PPy NT and the amino group of anti-cortisol IgG. The resulting field-effect transistor-type biosensor was utilized to evaluate various cortisol concentrations. Cortisol was sensitively measured to a detection limit of 2.7 × 10^−10^ M (100 pg/mL), with a dynamic range of 2.7 × 10^−10^ to 10^−7^ M; it exhibited rapid responses (<5 s). We believe that our approach can serve as an alternative to time-consuming and labor-intensive health questionnaires; it can also be used for diagnosis of underlying stress-related disorders.

## 1. Introduction

Cortisol is a critical glucocorticoid steroid hormone in humans, derived from cholesterol. Moreover, cortisol is a clinically proven stress biomarker that plays a vital role in the regulation of various physiological processes such as blood pressure, glucose levels, and carbohydrate metabolism [1,2]. Variations or abnormalities in physiological cortisol levels regulate acute and chronic stress responses; they also differentiate between disease states and thus serve as indicators of progression [3,4,5]. For example, excess cortisol levels contribute to the development of Cushing’s disease, including its symptoms of obesity, fatigue, and bone fragility [6]. In contrast, reduced cortisol levels lead to Addison’s disease, which is characterized by weight loss, darkening of skin folds, and fatigue [7]. Moreover, cortisol levels in body fluids demonstrate variability between men and women, as well as throughout each day. Nonetheless, these levels generally remain in the nanomolar range, with morning levels in the range of 138–690 nM and evening levels in the range of 55–386 nM [8,9]. Therefore, the ability to monitor cortisol levels in body fluid is critical in the diagnosis and monitoring of disease progression. Conventional methods to detect stress levels include interviews and counseling, self-questionnaires, electroencephalograms, and electrocardiograms. Recently, cortisol levels have been measured by radioimmunoassay, enzyme-linked immunosorbent assays, surface-enhanced Raman spectroscopy, ultraviolet spectroscopy, gas chromatography–mass spectrometry, and electrochemical sensors [10,11,12]. These techniques provide qualitative or quantitative determinations for use in diagnosis, although some result in bias [13]. Moreover, these techniques require extensive sample preparations, extended data analysis periods, and access to large instruments. However, a nanobiosensor platform can overcome these disadvantages; furthermore, it can rapidly analyze any physical state or target analyte [14,15,16].

Field-effect transistors (FETs) have attracted interest as primary candidates for the fabrication of state-of-the-art sensor platforms because they can achieve high current amplification with a relatively high signal-to-noise ratio. Importantly, one-dimensional nanomaterials that exhibit high charge carrier mobility along their long axes can be used in highly sensitive biosensors [17]. Among the one-dimensional nanomaterials available, the remarkable physical and chemical characteristics of one-dimensional conducting polymers (CPs) at the nanometer scale provide outstanding sensing performance in biosensor applications [18]. In particular, polypyrrole (PPy) is a biocompatible CP that is widely used in polymer nanotube (NT) FET biosensors [19]. 

Herein, we describe a high-performance PPy NT FET-type biosensor for stress hormone monitoring. The PPy NT was synthesized by the reverse cylindrical micelle method, then used as a transducer element in an FET-based cortisol biosensor. For selective cortisol detection, the cortisol antibody immunoglobulin G (IgG) was covalently functionalized onto the PPy NT surfaces. The fabricated sensor platform could recognize the target stress hormone, cortisol, with a limit of detection of 270 pM (100 pg/mL) and a dynamic range of 2.7 × 10^−10^ to 10^−7^ M. This liquid-ion gated FET system, based on the nano–bio interface, allowed us to achieve highly sensitive and specific detection of the target hormone, cortisol, from its analogs. 

## 2. Materials and Methods 

### 2.1. Material

Pyrrole (Py; 98%), pyrrole-2-carboxylic acid (P3CA, 99%), sodium bis(2-ethylhexyl)-sulfosuccinate (AOT; 96%), FeCl_3_ (97%), hexane (>95%), (3-aminopropyl)triethoxysilane (APTES; >98%), and 4-(4,6-dimethoxy-1,3,5-triazin-2-yl)-4-methylmorpholinium chloride (DMT-MM; >96%) were purchased from Sigma-Aldrich (MO, USA) and used without further purification. Cortisol (human; >98%), cortisone, prednisolone, and corticosterone were also purchased from Sigma-Aldrich. Mouse monoclonal anti-cortisol IgG was purchased from Abcam (CORT-2, Cat. No. ab1952, UK). Stock solutions of cortisol and its analogs were prepared in phosphate-buffered saline at 270 µM and stored at −20 °C. Working solutions were freshly prepared by dilution of the stock solution.

### 2.2. Synthesis of One-Dimensional PPy NT

The preparation of PPy NT was carried out using a reverse cylindrical micelle method [20]. AOT (15 mmol) was dissolved in hexane (40 mL) and stirred for 30 min to reach equilibrium. Then, aqueous FeCl_3_ (7 M, 1 mL) was added to the AOT/hexane cocktail to generate reverse cylindrical micelles containing cations. Py (3.75 mmol) and P3CA (0.25 mmol) were added dropwise into the reverse cylindrical micelle phase. The chemical oxidation polymerization of the Py and P3CA ([Py-COOH]/[Py] = 1:30) monomer “cocktail” proceeded for 3 h at 18 °C. The resulting (unmodified) carboxylated PPy NTs were thoroughly purified by washing with excess ethanol to remove the residual surfactant and reagents until a colorless solution with neutral pH appeared. The final products were obtained after drying under vacuum at room temperature for 24 h. Characterization of the synthesized carboxylated PPy NT was performed using Fourier transform infrared spectroscopy (FTIR; Alpha-p, Bruker, Germany), scanning electron microscopy (Magellan400, FEI, Hillsboro, OR, USA), X-ray photoelectron spectroscopy (PHI 5000, VersaProbe, Ulvac-PHI, Chigasaki, Japan), and a source meter (Keithley 2612A, Keithley, Cleveland, OH, USA).

### 2.3. Fabrication of PPyNT FET Sensor

An interdigitated microelectrode array (IDA) consisting of 80 pairs of gold electrodes was patterned on a glass substrate using an E-beam evaporator. The resulting IDA electrode had Au/Cr layers (100/10 nm in thickness), a width of 2 µm, a length of 2000 µm, and inter-electrode spacing of 2 µm. The surface of the electrode was cleaned using sonication in ethanol and deionized water. The electrode was finally dried under vacuum at room temperature for 1 h. A reaction vessel with a volume of 1 mL was designed and used for all solution-based measurements. 

The IDA electrode was treated with a 1 wt% APTES solution for 12 h to modify the Au/Cr-deposited glass surface with an amino-terminal group, then washed with distilled water. The surface of the electrode with the terminal group was functionalized with 3 wt% carboxylated PPy NT (40 µL, 1:30 ratio of P3CA:Py) and 1 wt% DMT-MM (40 µL) for 12 h. DMT-MM was used as a condensing agent to conjugate the carboxylic acids of the PPy NT and amines of the glass surface to their corresponding amides. The resulting IDA electrode was washed with distilled water several times. Subsequently, IgG (20 µg/mL) was immobilized onto the PPy NT/IDA electrode surface by means of the same condensing procedure. Finally, the IgG/PPy NT/IDA configuration was carefully washed with distilled water three times. This FET-type sensor platform, based on a liquid-ion gate, was connected to a source meter and computer to monitor its real-time response for stress hormone detection. 

### 2.4. Anti-Cortisol IgG/PPy NT/FET Based Cortisol Detection

The anti-cortisol IgG/PPy NT FET-type sensor was placed on the stage of a probe station (Model 4000, MS-Tech, Seoul, Korea), which was connected to the source meter [21]. Current–voltage (*I*–*V*) curves were measured by voltage scan (rate: 0.1 mV/s). The real-time response was obtained from its electrical signals, which were measured with the source meter. Changes in the current were normalized according to the equation
S(%) = ∆*I*/*I*_0_ = (*I* − *I*_0_)/*I*_0_(1)
where, *I*_0_ and *I* represent the initial and real-time currents, respectively. 

## 3. Results

### 3.1. Fabrication of Anti-Cortisol IgG/PPy NT FET Sensor

Among the family of CPs, PPy nanomaterials have been the most extensively investigated due to their unique properties, including a simple synthesis procedure, excellent electrical conductivity, high biocompatibility, and environmental stability [20,22]. In particular, PPy nanomaterials with different nanostructures and morphologies have been utilized as appropriate electrical channel elements for FET sensors [23]. The synthesis of PPy NTs involves two main steps, as illustrated in Figure 1a. PPy NTs, which have a tubular structure, were synthesized with the aid of cylindrical micelle templates in an apolar solvent. Copolymerization of Py with pyrrole-3-carboxylic acid on the surface of a cylindrical micelle yielded intrinsically functionalized PPy NTs [1]. This structure-guiding agent-based polymerization did not require a high temperature, strong acid, or strong base to remove the template after polymerization. 

Figure 1b depicts the fabrication procedure for the anti-cortisol IgG/PPy NT FET sensor platform. An IDA electrode was patterned on a glass substrate by conventional photolithography. The IDA consisted of a pair of gold electrodes with 80 fingers each, in which each part served as either the source (S) or drain (D) electrode. As shown in Figure 1b, PPy NTs were immobilized on the Au electrode surface through covalent bonds to sustain stable electrical contact between the PPy NT and Au electrode. To achieve this immobilization, the glass surface was first treated with APTES with an amine terminal group (Reaction 1). Then, PPy NT with a carboxyl group was immobilized by means of conjugation between the amine and carboxyl terminal groups (Reaction 2). Subsequently, anti-cortisol IgG containing a free-amine group was anchored on the PPy NT surface. The PPy NT was then chemically coupled with the anti-cortisol IgG by means of the same condensing reaction (Reaction 3). This anti-cortisol IgG-functionalized PPy NT FET system was used to measure stress levels by monitoring cortisol, which yielded a simple, accurate approach for new stress-related diagnostic methods. The detailed conjugation reactions of the anti-cortisol IgG/carboxylated PPy NT/amine-IDA electrode were as follows:**Reaction 1. hydrolysis and condensation**H_2_N(CH_2_)_3_Si(OCH_3_)-Substrate + 3H_2_O → H_2_N(CH_2_)_3_Si(OH)_3_ + 3CH_3_OHH_2_N(CH_2_)_3_Si(OH)_3_-Substrate + 3OH-Substrate → H_2_N(CH_2_)_3_Si(O)_3_-Substrate**Reaction 2. condensation**PPy-COOH + H_2_N(CH_2_)_3_Si(O)_3_-Substrate → PPy-CONH(CH_2_)_3_Si(O)_3_-Substrate**Reaction 3. condensation**PPy-COOH + H_2_N-IgG → PPy-CONH-IgG

### 3.2. Characterization of PPy NT FET Transducer

PPy NTs with uniform diameters were synthesized by a chemical oxidation polymerization process with bare pyrrole and carboxyl-pyrrole, then characterized by various analysis methods. Figure 2a displays SEM images of the FET system before and after conjugation of anti-cortisol IgG, respectively. The SEM images show that the bare PPy NT with a ~200 nm diameter was synthesized with very homogeneous surface morphologies (left image in Figure 2a) [24]. Functionalization of anti-cortisol IgG on the PPy NT surface was also confirmed (right image in Figure 2a). 

Raman spectroscopy and X-ray photoelectron spectroscopy (XPS) were used to characterize the synthesized PPy NTs. As shown in Figure 2b, the XPS spectra exhibited a narrow range of the C *1s* and N *1s* core levels, which demonstrates the change after anti-cortisol IgG immobilization on the PPy NT. The peaks of the C *1s* spectrum were assigned to four components that correspond to carbon atoms in different functional groups: the pyrrole ring C_1_ (C-C, 283.98 and 284.65 eV), C_2_ in C=N bonds (286.17 eV), C_3_ of a carboxyl group (O=C-O, 288.23 eV), and C_4_ of C-N and C-O bonds (290.20 eV; Appendix A) [25]. Based on the narrow spectrum of the C *1s*, the height of the carboxyl peaks was reduced by the amide bond between the carboxyl terminus of the PPy NT and the amino terminus of the IgG. The N *1s* core-level spectrum shows peaks of the N-C bond (399.72 eV) and -NH bond (398.07 eV) in the unmodified PPy NT. The peaks at 400.4 eV correspond to amide nitrogen (CO-NH, 400.69 eV), appearing after the surface modification by IgG (Appendix A) [26]. Hence, the C *1s* and N *1s* peaks clearly confirm that the anti-cortisol IgG was immobilized on the PPy NT surface.

Figure 2c shows the Raman spectra of PPy NT and anti-cortisol IgG/PPy NT. Notably, PPy NT has two major bands, at approximately 1600 and 1350 cm^−1^. The peak located at 1560–1620 cm^−1^ corresponds to the C=C backbone stretching of PPy and can be assigned mainly to the inter-ring C-C stretching vibration. The peak located at the lower frequency (1055 cm^−1^) corresponds to non-protonated PPy units; its intensity increases after deprotonation [27,28]. IgG, which is an antibody, is predominantly composed of α-helix (7%), β-sheet (47%), and other parts (i.e., rings and coils) [29,30]. After IgG conjugation to the PPy NT, the characteristic peaks are clearly visible; these represent distinctive secondary conformations of IgG. The predominant β-sheet structure in IgG can be identified by the characteristically higher amide I and II bands at approximately 1650 and 1350 cm^−1^. Typically, the amide I band is located at approximately 1672 cm^−1^, corresponding to the β-sheet structure, which is characteristic of IgG. However, the amide III region (1240–1350 cm^−1^) shows characteristics of an α-helix structure.

Figure 2d shows the Fourier Transform Infrared (FT-IR) spectra of PPy NT and anti-cortisol IgG/PPy NT. The characteristic absorption peaks associated with PPy can be observed at 3400, 1549, 1424, 1046, 968, and 794 cm^−1^, corresponding to N-H, C=C, C-C, and C-N stretching vibrations, as well as C-H in-plane and C-H out-of-plane vibrations, respectively [31]. After the conjugation of anti-cortisol IgG to PPy NT, its FTIR spectrum was investigated to confirm the presence of IgG. The major characteristic peaks correspond to the presence of various oxygen-containing functional groups at approximately 3400 cm^−1^ (-OH stretching of the hydroxyl group). In addition, amide I and amide II bands in the FTIR spectrum are two significant bands for protein, which consist of peptide groups with structural repeating units. The amide I and II bands are mainly associated with C=O stretching (1600–1700 cm^−1^) and N-H bending vibrations, respectively. These FTIR spectra were in good agreement with the previous results, confirming the presence of PPy NT and protein [32]. Therefore, the FTIR spectra of the prepared PPy NT and anti-cortisol IgG/PPy NT clearly show the synthesis of PPy NT and the structure-related details of its expected composition.

### 3.3. Sensing Behavior of Liquid-Ion Gated Anti-Cortisol IgG/PPy NT FET

To evaluate the performance of our FET configuration, the liquid-ion gate system was constructed with the anti-cortisol IgG/PPy NT biosensor in phosphate-buffered saline (pH 7.4). Figure 3a shows a schematic representation of the liquid-ion gated FET biosensor, consisting of two Au electrodes (S and D). The electrical properties of PPy NTs and anti-cortisol IgG/PPy NTs were investigated by measuring their *I–V* curves. As shown in Figure 3b, linear *I–V* curves were observed over a range of −2.5 to +2.5 V, indicating that Ohmic connections were formed between the IgG/PPy NTs and the Au electrodes. The conductivity of unmodified PPy NTs was higher than that of IgG-modified PPy NTs, which is caused by increased resistance via the IgG attachments on the surfaces of PPy NT. The Ohmic contact was maintained after the coupling and washing steps in the surface functionalization procedures. These results implied that the covalent immobilization of IgG and PPy NT on the Au electrode yielded a reliable electrical connection.

Figure 3c depicts the transfer characteristics (*I*_ds_*–V*_g_, where *I*_ds_ is the source–drain current and *V*_g_ is the gate voltage) of our FET-type biosensor device following treatment and immobilization of IgG. To measure the transfer characteristics of the FET sensor, a constant bias voltage *V*_ds_ of −1 mV was applied across the S and D electrodes. *V*_g_ was applied by immersing the Pt reference electrode in the phosphate-buffered saline electrolyte on the top of the PPy NT FET sensor. The hysteresis was tested at the fixed gate voltage range, the current gap at the different direction sweep was observed by Δ*I*_ds_ = 0.1 mA at *V*_g_ = −0.6 V (Appendix A) [33]. Upon conjugation of IgG with the PPy NT surface, the transfer curve shifted. Normally, an antibody is an amphoteric electrolyte composed of many carboxyl and amino bases. The isoelectric point (pI) of IgG, including its anti-cortisol antibody, is approximately pH 5.0–5.5 [10]. Hence, the charge state of the IgG became negative at pH 7.4 in our measurements [11]. Although the PPy NT FET sensor changed the current, it maintained typical p-type semiconductor characteristics. This result clearly indicated that the fabricated PPy NT FET device was able to sensitively detect the modulation of charges at the surface of its PPy NTs, in addition to reflecting significant changes in the FET channel in terms of the current–voltage responses. Based on these observations, the interactions of PPy NTs after each modification step, as well as the responses to varying concentrations of cortisol, can be evaluated by measuring the changes in current–voltage characteristics. 

To investigate the electrical characteristics of PPy NTs as conductive channels, we measured the *I*_ds_*–V*_ds_ characteristics of our liquid-ion gated FET platform (Figure 3d). Typically, the liquid-ion gate configuration in an FET system enables amplification of the current due to binding events on the surface of a nanomaterial by applying *V*_g_ via its gate electrode. In our test, *V*_g_ was varied in the range of 0.4 to −1.2 V in steps of 0.2 V, with a sweep rate of 0.2 V/s. When a more negative gate bias was applied, *I*_ds_ increased (i.e., became more negative). The negatively increasing *V*_g_ led to an increasing *I*_ds_ current, because a negative charge applied to the surface of an IgG/PPy NT changes the state of the intrinsic oxidation level in the PPy chains [34]. This result showed that the liquid-ion gated FET system has a clearly defined p-type transistor characteristic, with holes as the primary charge carrier [21]. In addition, the *I*_ds_*–V*_g_ characteristic depending on pH variation was observed in the range of pH 4.8, 7.4, and 8.8, and the charge density was changed by PPy NT surface charge variation (Appendix A). Therefore, binding events on the surface of PPy NT can be monitored at different cortisol concentrations by measuring real-time current changes under controlled gate voltages.

### 3.4. Real-Time Response to Cortisol

To evaluate the sensing performance of our FET-type biosensor, we measured its electrical responses in the liquid-ion gated configuration. As mentioned previously, the anti-cortisol IgG was immobilized onto the PPy NT surface, which induces electrical signals after selective binding with cortisol. The dependence of *I_ds_* on various cortisol concentrations was measured under the following conditions: *V*_ds_ = −1 mV, *V*_g_ = −100 mV. The sensitivity was determined by changes in the normalized current, as calculated by Equation (1). The optimal concentration of the anti-cortisol IgG was determined to be 20 μg/mL (Appendix A). As shown in Figure 4a, significant current changes were observed as cortisol concentrations increased. Upon sequential addition of various concentrations, the cortisol was measured with a detection limit of 2.7 × 10^−10^ M (100 pg/mL), a dynamic range of 2.7 × 10^−10^ to 10^−7^ M, and rapid responses (<5 s) (the inset of Figure 4a) [35]. In addition, the sensor life span test was performed for 7 days and the sensor performance was maintained over 50% compared with its initial sensitivity (Appendix A). These results imply that the binding events between varying concentrations of cortisol and anti-cortisol IgG can directly affect the charge carrier density on the surface of the PPy NT by means of its liquid-ion gate. More specifically, this was caused by the accumulation of hole-type charge carriers. In addition, the sensing performance can further improved by controlling the Debye length (λ_D_). For example, Jang et al. have described an antibody embedded polymer matrix configuration in FET sensors that allows the effective detection of charge variations between membrane–substrate interfaces [36]. Figure 4b shows the corresponding calibration curve of our cortisol FET sensor, which demonstrates a linear response. 

To achieve specific cortisol sensing with the liquid-ion gated FET sensor, various hormones comprising interference substances were prepared, as shown in Figure 4c. Cortisol, cortisone, corticosterone, and prednisolone have similarities in terms of molecular structure and coexist in the in vivo environment. The specificity of our sensor platform toward cortisol was tested by applying these substances (Figure 4d). The anti-cortisol IgG-based FET assay showed a current change when cortisol was injected at 270 pM. In contrast, no significant responses were observed with high concentrations of other structurally related hormones. Thus, a specific response for cortisol was observed at 2.7 × 10^−10^ M in the presence of a series of similarly structured hormone analogs. Based on this result, our sensor demonstrated a reliable and reproducible sensing performance for the quantification and detection of cortisol.

## 4. Conclusions

We described a novel approach for measuring cortisol levels based on an anti-cortisol IgG/PPy NT FET-type biosensor. Levels of cortisol in blood or saliva can be measured and used as a robust indicator of stress level. The resulting FET-type biosensor was utilized to measure cortisol. Cortisol was sensitively and selectively measured to limit of detection (LOD) of 2.7 × 10^−10^ M, with a dynamic range of 2.7 × 10^−10^ to 10^−7^ M (response time: <5 s). This novel approach could be used to develop targeted specific medications for stress-related disorders.

## Figures and Tables

**Figure 1 micromachines-11-00439-f001:**
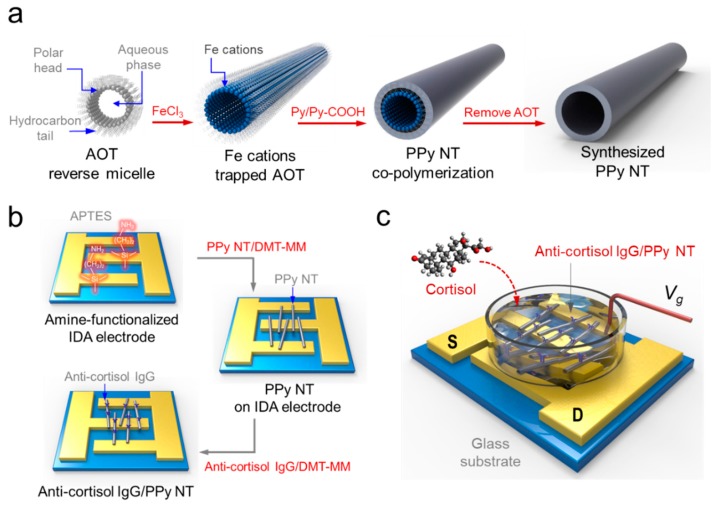
Fabrication of anti-cortisol immunoglobulin G (IgG)/polypyrrole (PPy) nanotube (NT) field-effect transistor (FET)-type biosensor for stress hormone detection. (**a**) Synthesis of PPy NT by reverse cylindrical micelle method. (**b**) Schematic illustration of the fabrication procedure for anti-cortisol IgG/PPy NT FET configuration. (**c**) (S) and (D) represent source and drain electrodes, respectively. The FET sensor system consists of three electrodes that were immersed in phosphate-buffered saline buffer (pH 7.4) as a liquid-ion gate. The current flows from *V*_ds_ to *I*_ds_.

**Figure 2 micromachines-11-00439-f002:**
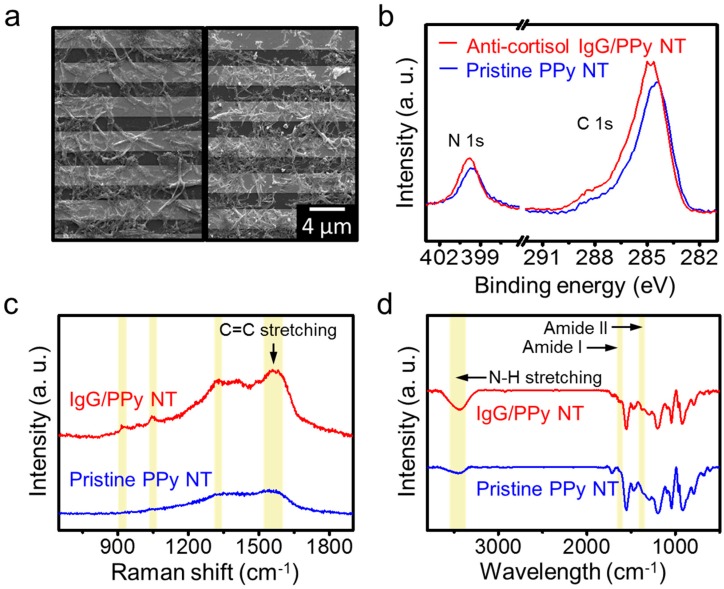
Characteristics of synthesized, functionalized polypyrrole (PPy) nanotubes (NTs). (**a**) SEM images before (i) and after (ii) conjugation of anti-cortisol IgG onto the surface of PPy NTs. PPy NTs were immobilized on the Au fingers of the IDA electrode, and their surfaces were modified with IgG using a condensing agent, DMT-MM. (**b**) X-ray photoelectron spectroscopy (XPS) spectra, (**c**) Raman spectra, and (**d**) FT-IR analyses of unmodified PPy NTs and anti-cortisol IgG-modified PPy NTs.

**Figure 3 micromachines-11-00439-f003:**
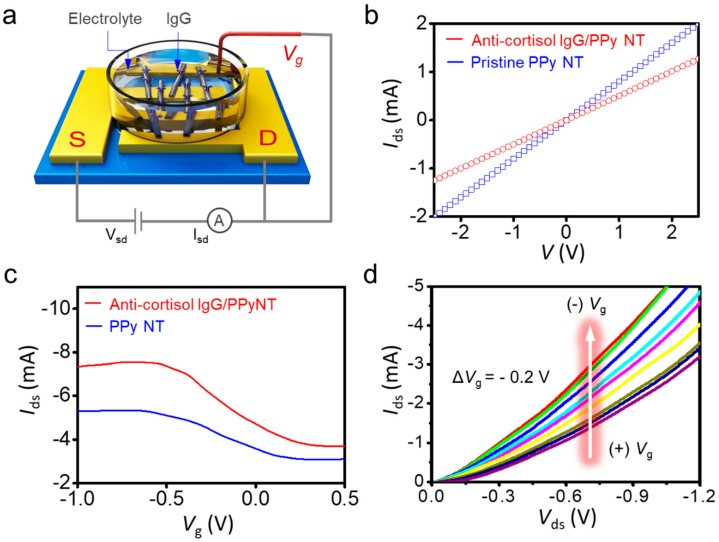
Electrical characteristics of the anti-cortisol IgG/PPyNT FET configuration. (**a**) Schematic representation of liquid-ion gated FET system (S: source; D: drain; V_g_: gating voltage). (**b**) *I-V* values of before and after the conjugation of anti-cortisol IgG. (**c**) Transfer curves. (**d**) Source–drain current (*I*_ds_)–bias voltage (*V*_ds_) curves from the anti-cortisol IgG FET configuration, with various gate voltages from 0.4 to −1.2 V.

**Figure 4 micromachines-11-00439-f004:**
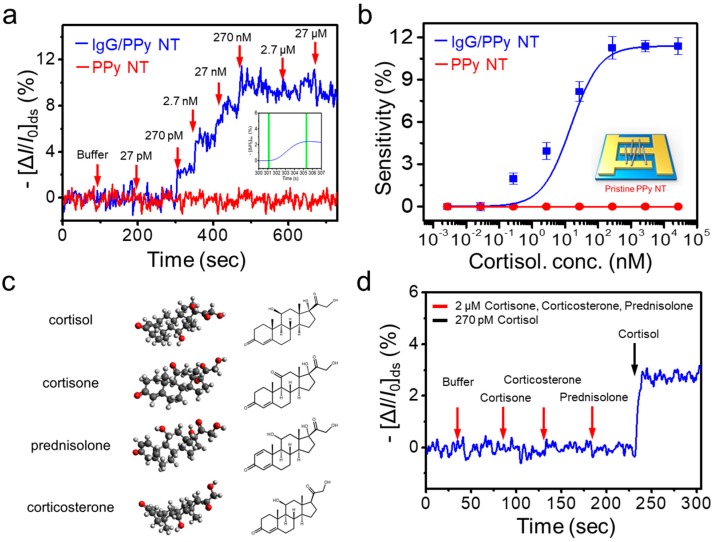
Real-time responses of the anti-cortisol IgG/PPyNT FET system. (**a**) Real-time responses for sensitivity at increasing cortisol concentrations. (**b**) Dose-dependent responses of cortisol in the range of 27 pM to 27 µM (n = 5). (**c**) Chemical structures of the target stress hormone (cortisol) and non-target hormones (cortisone, corticosterone, and prednisolone). (**d**) Real-time response for specificity among high concentrations of diverse molecules with similar structures. The concentrations of cortisol and non-target hormones are 270 pM and 2 μM, respectively.

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
