# Peer review of "Ultrasensitive Stress Biomarker Detection Using Polypyrrole Nanotube Coupled to a Field-Effect Transistor"

_micromachines, 2020, doi:10.3390/mi11040439_

Round 1

Reviewer 1 Report

The innovation of this manuscript is to investigate a FET sensor coupled with polypyrrole nanotube for the detection of cortisol. After careful reading of this manuscript, I have the following comment:

  • Life span test of the sensor should be conducted.
  • A study of the optimization of the concentration of anti-cortisol immunoglobulin G should be conducted.
  • The effect of the pH of the test buffer on the sensor performance should be conducted.
  • Comparison with other sensors for cortisol detection should be made.
    • Jang, H.-J.; Lee, T.; Song, J.; Russell, L.; Li, H.; Dailey, J.; Searson, P.C.; Katz, H.E. Electronic Cortisol Detection Using an Antibody-Embedded Polymer Coupled to a Field-Effect Transistor. ACS Appl. Mater. Interfaces 2018, 10, 16233–16237.
  • In figure 4(d) caption, I think it should be 270pM rather than 27 pM cortisol injection.
  • In line272 and 273, authors said that “The anti-cortisol IgG-based FET assay showed a current change when cortisol was injected at 27 pM.” I don’t agree as a response is happen in 270pM.
  • What is the concentration of the cortisone, corticosterone, and prednisolone?
  • A hysteresis interference test should be conducted.

Author Response

Comment 1.

Life span test of the sensor should be conducted.

Response

We appreciate the reviewer for the great comment. As the reviewer suggested, we have performed the life span test with our sensor platform and have added new results in Figure S6 and related sentence on page 8, line 269.

 “In addition, the sensor life span test performed for 7 days, and the sensor performance maintained over 50 % from comparison with initial sensitivity (Figure S6).”

Please see attachment file for detail response and figure

Comment 2.

A study of the optimization of the concentration of anti-cortisol immunoglobulin G should be conducted.

Response

Thank you for the reviewer’s valuable comment. The anti-cortisol IgG was immobilized and evaluated on the PPy NT at each different concentration (10, 20, and 40 µg/mL) to figure out their optimal concentration. The resulting optimal concentration of IgG was determined to be 20 µg/ml from our results.

We have added Figure S4 and the following sentence on page 8, line 265.

“The optimal concentration of the anti-cortisol IgG was determined to be 20 μg/ml (Figure S5).” 

Figure S5. The real-time response of depending on the anti-cortisol concentrations (10, 20, and 40 µg/mL).

Please see attachment file for detail response and figure

Comment 3.

The effect of the pH of the test buffer on the sensor performance should be conducted.

Response

We appreciate the reviewer’s comment. As the reviewer’s suggested, we obtained the transfer curve characteristic of depending on pH variation (Vds = - 1 mV). The shift of transfer curve was related with an isoelectric point (pI), which was mentioned in line 229. The anti-cortisol IgG has a positive charge at the state below pI; therefore, the transfer characteristic exhibited decreasing hole-type charge carrier.

We have added Figure S4 and the following sentence on page 7, line 248.

  • “In addition, the Ids–Vg characteristic depending on pH variation was observed in range pH 4.8, 7.4 and 8.8, and the charge density was changed by PPy NT surface charge variation (Figure S4).”

Figure S4. The transfer curve of anti-cortisol IgG/PPyNT FET regarding pH effect (pH 4.8, 7.4, and 8.8).

Please see attachment file for detail response and figure

Comment 4.

Comparison with other sensors for cortisol detection should be made.

Jang, H.-J.; Lee, T.; Song, J.; Russell, L.; Li, H.; Dailey, J.; Searson, P.C.; Katz, H.E. Electronic Cortisol Detection Using an Antibody-Embedded Polymer Coupled to a Field-Effect Transistor. ACS Appl. Mater. Interfaces 2018, 10, 16233–16237.

Response

We appreciate the reviewer’s comment. As the reviewer’s suggested, we have added the description as a suggestion to improve the sensing performance of our platform in comparison to the above-recommended reference. The following sentence with a new reference was newly added in page 8, line 274.

  • “In addition, the sensing performance can further improve by controlling the Debye length (λD). For example, Jang et al., describes the antibody embedded polymer matrix configuration in a FET sensor, which allows to effectively detect the charge variations between the membrane-substrate interfaces.”
  • Jang, H.-J.; Lee, T.; Song, J.; Russell, L.; Li, H.; Dailey, J.; Searson, P.C.; Katz, H.E. Electronic Cortisol Detection Using an Antibody-Embedded Polymer Coupled to a Field-Effect Transistor. ACS Appl. Mater. Interfaces 2018, 10, 16233–16237.

Comment 5.

In figure 4(d) caption, I think it should be 270pM rather than 27 pM cortisol injection.

Response

Thank you for your comment. We have corrected it to 270 pM on page 8, line 284.

Comment 6.

In line 272 and 273, authors said that “The anti-cortisol IgG-based FET assay showed a current change when cortisol was injected at 27 pM.” I don’t agree as a response is happen in 270 pM.

What is the concentration of the cortisone, corticosterone, and prednisolone?

Response

We appreciate the reviewer’s critical comment. The concentration of cortisol in Fig 4d was 270 pM, however, the concentrations of the injected interference substances were 2 μM. The caption of Figure 4d was revised as follows on page 9, line 294.

In the revised manuscript,

“The concentrations of cortisol and non-target hormones were 270 pM and 2 μM, respectively.”

Comment 7.

A hysteresis interference test should be conducted.

Response

Thank you for the reviewer’s comment. As the reviewer’s suggested, we have performed a hysteresis test, which was measured from the forward and reverse sweeps of Ids versus Vg. The different current value was confirmed at the fixed gate voltage range, the maximum difference of current was calculated as ΔIds = 0.1 mA at - 0.6 V of gate voltage.

We have added Figure S3, new reference, and the following sentence on page 6, line 226.

  • “The hysteresis was tested at the fixed gate voltage range, the current gap at the different direction sweep was observed by ΔIds = 0.1 mA at Vg = - 0.6 V (Figure S3)”
  • Park, R.S.; Shulaker, M.M.; Hills, G.; Suriyasena Liyanage, L.; Lee, S.; Tang, A.; Mitra, S.; Wong, H.S.P. Hysteresis in Carbon Nanotube Transistors: Measurement and Analysis of Trap Density, Energy Level, and Spatial Distribution. ACS Nano, 2016, 10, 4599–4608.

 Figure S3. Typical transfer curve for hysteresis confirmation measured at Vds = - 1 mV and between the cyclic sweeps.

Please see attachment file for detail response and figure

Reviewer 2 Report

In this study, cortisol biomarker based on a conducting polymer nanotube field-effect transistor platform with low detection limit of 2.7 × 10-10 M has been provided. However, there are several unclear description which should be further address. Hence, I would suggest acceptance after minor revision. The following are the missing parts that should be further clarified.

  1. In figure 3b, the conductivity of unmodified PPy NTs was higher than that of IgG-modified PPy NTs. The author should provide the mechanism behind it.
  2. Response time is the essential factor to evaluate the performance of sensor. The real-time response should be enlarged to address the rapid response claimed in the manuscript.
  3. In figure 4b, sensitivity should be defined in the experiment first.
  4. The structure of anti-cortisol IgG should be provided in figure 4 as well.
  5. The sensitivity of cortisol in blood or saliva may be different in the case of water solution. Besides, the sentence in line 287 mentioned “Salivary cortisol and free cortisol in blood are more robust indicators of stress, compared to total blood cortisol.” which is not relative to the results in the manuscript. Authors should revised it.

Author Response

Comment 1.

In Figure 3b, the conductivity of unmodified PPy NTs was higher than that of IgG-modified PPy NTs. The author should provide the mechanism behind it.

Response

Thank you for the reviewer’s comment. In our test, the conductivity was decreased by as attachment of the anti-cortisol antibody on the PPy NT surface. The detailed explanation and mechanism were already stated in the manuscript on page 6, line 217. Also, we have modified the sentence to provide more statements on page 6, line 217.  

“The conductivity of unmodified PPy NTs was higher than that of IgG-modified PPy NTs, which is caused by increased resistance via the IgG attachment on the surface of PPy NT.”

Comment 2.

Response time is the essential factor to evaluate the performance of sensor. The real-time response should be enlarged to address the rapid response claimed in the manuscript.

Response

Thank you for the reviewer’s comment. The enlarged response time was inserted in figure 4a, and was confirmed below 5 sec (ca. 4 sec). The new reference and sentence added on page 8, line 269.

“(the inset of Figure 4a)”

Please see attachment file for detail response and figure

Comment 3.

In figure 4b, sensitivity should be defined in the experiment first.

Response

In accordance with the reviewer’s suggestion, the following sentence for sensitivity calculation was added on page 8, line 264. 

“The sensitivity was determined by changes in the normalized current, as calculated by Equation (1).”

Comment 4.

The structure of anti-cortisol IgG should be provided in figure 4 as well.

Response

Thank you for the reviewer’s comment. As the reviewer suggested, we have added the enlarged schematic drawing for anti-cortisol IgG in Figure 4b.

Please see attachment file for detail response and figure

Comment 5.

The sensitivity of cortisol in blood or saliva may be different in the case of water solution. Besides, the sentence in line 287 mentioned “Salivary cortisol and free cortisol in blood are more robust indicators of stress, compared to total blood cortisol.” which is not relative to the results in the manuscript. Authors should revised it.

Response

We appreciate the reviewer’s valuable comment. We have deleted that sentence and have revised the Conclusion section on page 9, line 299.

“We described a novel approach for measurement of cortisol level based on an anti-cortisol IgG/PPy NT FET-type biosensor. Levels of cortisol in blood or saliva can be measured and used as a robust indicator of stress level. Salivary cortisol and free cortisol in the blood are more robust indicators of stress, compared to total blood cortisol. The resulting FET-type biosensor was utilized to measure cortisol. Cortisol was sensitively and selectively measured to a LOD of 2.7 × 10-10 M, with a dynamic range of 2.7 × 10-10 to 10-8 M (response time: < 5 sec). This novel approach could be used to develop targeted specific medications for stress-related disorders.”

Round 2

Reviewer 1 Report

Authors tried to answer all the reviewer's questions and this revised manuscript is good enough to provide adequate useful information to reader.